# Fibrotic Changes to Schlemm’s Canal Endothelial Cells in Glaucoma

**DOI:** 10.3390/ijms22179446

**Published:** 2021-08-31

**Authors:** Ruth A. Kelly, Kristin M. Perkumas, Matthew Campbell, G. Jane Farrar, W. Daniel Stamer, Pete Humphries, Jeffrey O’Callaghan, Colm J. O’Brien

**Affiliations:** 1Ocular Genetics Unit, Smurfit Institute of Genetics, Trinity College Dublin, D02 PN40 Dublin, Ireland; jane.farrar@tcd.ie (G.J.F.); pete.humphries@tcd.ie (P.H.); ocallaje@tcd.ie (J.O.); 2Department of Ophthalmology, Duke University Eye Center, Durham, NC 27705, USA; kristin.perkumas@duke.edu (K.M.P.); william.stamer@duke.edu (W.D.S.); 3Neurovascular Research Laboratory, Smurfit Institute of Genetics, Trinity College Dublin, D02 PN40 Dublin, Ireland; campbem2@tcd.ie; 4Department of Ophthalmology, Mater Misericordiae University Hospital, D07 AX57 Dublin, Ireland; cobrien@mater.ie

**Keywords:** glaucoma, fibrosis, Schlemm’s canal, endothelial cells, endothelial–mesenchymal transition, proliferation

## Abstract

Previous studies have shown that glaucomatous Schlemm’s canal endothelial cells (gSCECs) are stiffer and associated with reduced porosity and increased extracellular matrix (ECM) material compared to SCECs from healthy individuals. We hypothesised that Schlemm’s canal (SC) cell stiffening was a function of fibrotic changes occurring at the inner wall of SC in glaucoma. This study was performed in primary cell cultures isolated from the SC lumen of human donor eyes. RNA and protein quantification of both fibrotic and endothelial cell markers was carried out on both healthy and gSCECs. Functional assays to assess cell density, size, migration, proliferation, and mitochondrial function of these cells were also carried out. Indeed, we found that gSCECs deviate from typical endothelial cell characteristics and exhibit a more fibrotic phenotype. For example, gSCECs expressed significantly higher protein levels of the fibrotic markers α-SMA, collagen I-α1, and fibronectin, as well as significantly increased protein expression of TGFβ-2, the main driver of fibrosis, compared to healthy SCECs. Interestingly, we observed a significant increase in protein expression of endothelial marker VE-cadherin in gSCECs, compared to healthy SCECs. gSCECs also appeared to be significantly larger, and surprisingly proliferate and migrate at a significantly higher rate, as well as showing significantly reduced mitochondrial activity, compared to healthy SCECs.

## 1. Introduction

Glaucoma is the leading cause of blindness worldwide, after cataracts. It is expected that the global prevalence of the disease will increase dramatically to approximately 111.8 million by 2040 [1]. The pathology of glaucoma is not fully understood; however, effective lowering of intraocular pressure (IOP) slows progression [2]. IOP can be modified by therapeutically targeting the production or drainage of aqueous humour (AH) [3]. The primary route of drainage and site responsible for IOP regulation is the conventional outflow pathway, which is mainly comprised of the trabecular meshwork (TM) and Schlemm’s canal (SC) [4]. AH first filters through the TM, then across the inner wall of Schlemm’s canal via micrometre-sized pores that form in giant vacuoles or junctions between SC cells themselves. Once in the SC lumen, AH is directed towards collector channels and aqueous veins that empty into the venous system [3,5,6]. As AH funnels from the wider juxtacanalicular tissue (JCT) region towards the narrow pores of the inner wall of SC, AH outflow resistance increases. The precise mechanisms by which AH resistance is generated are unclear, but it definitely occurs near the inner wall of SC [5].

Pore density of the inner wall of SC negatively correlates with outflow resistance, with glaucomatous SC having fewer pores [7]. Similarly, glaucomatous SCECs (gSCECs) form fewer pores than healthy SCECs [8,9]. gSCECs also appear to be stiffer than cells from a healthy individual, which may explain the decreased pore-forming ability [5,6]. It is of interest to note that a study carried out by Cai et al. (2015) showed 113 genes with at least twofold expression changes between gSCECs and SCECs [10]. These gene changes are involved in several pathways, including cell adhesion, heparin binding, glycosaminoglycan binding, and ECM, all of which may contribute to cell stiffness and decreased pore formation in gSCECs, potentially causing increased outflow resistance and elevated IOP [10].

One of the major driving forces of cell stiffening/fibrosis is TGF-β2, which is present at elevated levels in glaucomatous AH [11,12,13,14,15]. Fibrosis is the disruption of the normal structural components that form tissue, involving the accumulation of extracellular matrix (ECM) proteins, which results in the formation of non-functional scar tissue [16]. Some assert that progressive fibrosis along the outflow pathway, specifically the TM, accounts for most of the damage in glaucoma pathology [17]. Characteristic features of fibrosis found in glaucomatous TM regions include increased ECM, increased expression of TGF-β, as well as increased expression of mesenchymal markers, such as α-smooth muscle actin (α-SMA) [18,19]. Several in vitro studies have shown that TM cells treated with TGF-β2 have elevated expression of collagens, fibronectin, actin stress fibres, thrombospondin-1, lysyl oxidase, transglutaminase, and plasminogen activator inhibitor-1 (PAI-1) [20,21,22,23,24]. Results showed that changes in ECM material, as well as alterations to cytoskeletal proteins, occur in these cells following treatment with TGF-β2 [25]. It has also been shown that perfusing ex vivo anterior segments with TGF-β2, for approximately one week, reduced the outflow facility by 27% and affected the ECM structure of the TM [26]. Many of these studies have linked the activity of TGF-β2 to the upregulation of ECM deposition, in turn reducing outflow facility and increasing IOP [20,26].

The expression of the fibrotic marker α-SMA in TM cells is a characteristic in the differentiation of a greater proportion of these TM cells becoming myofibroblasts [19]. Tissue damage triggers the transformation of dormant tissue-resident cells into a mesenchymal-like cell phenotype, called myofibroblasts. This process involves the formation of α-SMA stress fibres, which are an abundant feature of glaucomatous TM tissue [27,28]. It has also been shown that epithelial–mesenchymal transition (EMT) can occur in TM cells by exposing them in vitro to collagen-I, fibronectin, and laminin resulting in the dissociation of cell–cell contact, elongation of actin stress fibres, and expression of mesenchymal markers, such as fibronectin and α-SMA [18]. However, since TM cells are not epithelial in nature, the myofibroblast trans-differentiation cannot be classified as EMT in the strictest sense but has been described to display EMT-like features, due to the loss of cell–cell contact, mesenchymal-like phenotype, increase in cell motility, upregulation of ECM materials and phosphorylation of TGF-β signalling molecules [18].

Endothelial–mesenchymal transition (EndMT) is a variant of EMT but involves endothelial cells. Exposing such cells to TGF-β results in the progression of EndMT, resulting in upregulation of α-SMA and abnormal ECM secretion [29]. The transition of healthy endothelial cells into mesenchymal-like state results in the loss of healthy endothelial properties such as pore formation and, in turn, a gain of fibrotic-like characteristics, including expression of fibrotic markers, increased cell stiffness, and increased ECM stiffness [30]. TGF-β2 levels have been shown to be increased in AH of glaucoma patients [12,31]. We hypothesise that TGF-β2-induced EndMT is occurring in endothelial cells of the inner wall of SC, in parallel with changes that are known to occur in the TM [19,32,33].

## 2. Results

### 2.1. Fibrotic Marker Expression in Healthy versus Glaucomatous SCECs

Relative mRNA transcript (left) and protein (right) expression of fibrotic markers α-SMA, collagen I-α1, and fibronectin are shown below, as well as pro-fibrotic marker TGF-β2 (Figure 1). This study was performed in primary cell cultures isolated from the SC lumen of human donor eyes. Biological replicates (*n* = 5 for SCEC, specifically SCEC 71, 91a, 89, 69, and 74 and *n* = 3 for gSCEC, specifically gSCEC 57, 64, and 90) are shown below, and statistical analysis was carried out using an unpaired Student’s *t*-test.

A significant increase in α-SMA transcript (*p* = 0.0003) and protein (*p* = 0.0105) expression of 4.1 (2.8, 5.4) and 2.4 (0.8, 4.0) fold, respectively, for gSCECs, compared to SCECs, was observed (mean (CI)). Similarly, a significant increase in TGF-β2 transcript (*p* = 0.0431) and protein (*p* = 0.0389) expression of 3.7 (0.2, 7.3) and 1.7 (0.1, 3.3) fold, respectively, for gSCECs, compared to SCECs, was recorded. Excluding outlier SCEC 91a, no change to statistical significance of TGF-β2 transcript expression was observed (*p* = 0.0182). While collagen I-α1 transcript was not significantly different (fold change of 0.1 (−0.7, 1.0), *p* = 0.7630), protein expression was significantly upregulated (4.7 (1.1, 8.4) fold change *p* = 0.0190) in gSCECs, compared to SCECs. With fibronectin, its transcript levels were not different (0.7 (−1.9, 3.3), *p* = 0.5314); however, protein expression significantly increased by 2.3 (0.1, 4.6) fold (*p* = 0.0457) for gSCECs, compared to SCECs.

### 2.2. Comparison of Endothelial Cell Marker Expression in Healthy versus Glaucomatous SCECs

Relative mRNA transcript expression and protein expression of the endothelial cell markers, vascular endothelial cadherin (VE-cadherin), vinculin, and von Willebrand factor (vWF) were analysed (Figure 2). Biological replicates (*n* = 5 for SCEC, specifically SCEC 71, 91a, 89, 69, and 74, and *n* = 3 for gSCEC, specifically gSCEC 57, 64, and 90) are shown, and statistical analysis was carried out using an unpaired Student’s *t*-test.

There was no significant difference in VE-cadherin transcript expression (mean fold change (CI); 1.0 (−0.4, 2.4), *p* = 0.1303); however, protein expression was significantly greater by 2.5 fold (0.8, 4.3) (*p* = 0.0115) in gSCECs, compared to SCECs. No significant difference in vinculin transcript and protein expression (0.4 (−1.0, 1.7) fold change, *p* = 0.5111 and 0.3 (−0.2, 0.8) fold change, *p* = 0.1727, respectively) was observed in gSCECs, compared to SCECs. Likewise, we observed no significant difference in vWF transcript and protein expression (0.6 (−0.5, 1.8) fold change, *p* = 0.2079 and 0.1 (−0.4, 0.5) fold change, *p* = 0.7368, respectively) in gSCECs, compared to SCECs.

### 2.3. Elevated Levels of α-SMA and F-Actin Protein in gSCECs, Compared to Healthy SCECs

Both α-SMA and filamentous actin (F-actin) protein expression were visualised using immunocytochemistry (ICC) (Figure 3). An increase in α-SMA protein expression was observed in gSCECs (Figure 3C), compared to SCECs (Figure 3A), which corresponds with protein quantification data (Figure 1). Increased expression of α-SMA was prominent, particularly enveloping the nucleus of gSCECs (Figure 3C). These gSCECs also appeared to be larger in size, expressing more cytoskeletal actin protein (Figure 3C), compared to healthy SCECs (Figure 3A). F-actin cytoskeletal protein expression also appeared to be increased, with an increase in actin stress fibres, in gSCECs (Figure 3D), compared to SCECs (Figure 3B). Importantly, these cells were seeded at the same time and at the same density; however, gSCECs appeared more confluent (Figure 3C,D) than SCECs (Figure 3A,B). SCEC 89 and gSCEC 90 are shown below in Figure 3.

### 2.4. Increased Cell Size, Proliferation, Migration and Reduced Mitochondrial Activity in Glaucomatous SCECs, Compared to Healthy SCECs

Throughout the course of these in vitro experiments, gSCECs appeared to be growing at a faster rate, compared to healthy SCECs. A number of functional and morphological experiments were carried out on both healthy SCECs and gSCECs to investigate further apparent differences in proliferation. After identical seeding densities, there was a significant increase of 1.5 × 10^6^ (7.1 × 10^4^, 2.9 × 10^6^) cells/mL (mean (CI), *p* = 0.0417, unpaired *t*-test, *n* = 8 for SCEC, specifically SCEC 71, 91a, 89, 69, 74, 68, 81, and 86, and *n* = 3 for gSCEC, specifically gSCEC 57, 64, and 90) observed in gSCECs, compared to SCECs (Figure 4A) after 4 days of culture. This finding was interesting given that the average cell diameter of gSCECs seeded in culture media tended to be larger (mean cell diameter increase of 3.9 (−2.7, 10.5) μm, *p* = 0.1941, unpaired *t*-test, *n* = 5 for SCEC, specifically SCEC 81, 91a, 89, 74, and 71, and *n* = 3 for gSCEC, specifically gSCEC 57, 64, and 90) but was not statistically significant, using the LUNA Automated Cell Counter (Figure 4B). Using a second method, the area of both SCEC and gSCEC seeded onto a chamber slide was calculated using confocal microscopy and ZEN software. A significant increase in mean cell area of 47.9 (4.3, 91.5) μm^2^ (*p* = 0.0361, unpaired *t*-test, *n* = 5 for SCEC, specifically SCEC 81, 91a, 89, 74, and 71 and *n* = 3 for gSCEC, specifically gSCEC 57, 64, and 90) was observed for gSCECs, compared to SCECs (Figure 4C). A third method (MTS assay) was used to evaluate the proliferation of both SCECs and gSCECs. In line with previous results, we observed a significant increase in relative absorbance of 0.8 (0.7, 1.0) nm (*p* < 0.0001, unpaired *t*-test, *n* = 4 for SCEC, specifically SCEC 91a, 89, 69, and 74, and *n* = 3 for gSCEC, specifically gSCEC 57, 64, and 90) in gSCECs, compared to SCECs, just 24 h post seeding (Figure 4D).

The ability to repair a damaged inner wall of SC is an important feature to maintain an intact blood–aqueous barrier. We modelled mechanically induced inner wall disruption using a scratch test to investigate cellular mobility and proliferation rates. Consistent with their elevated doubling time, gSCECs showed a significant increase of 0.3 (0.1, 0.5) fold change of scratched area (*p* = 0.0054, unpaired *t*-test, *n* = 5 for SCEC, specifically SCEC 71, 91a, 89, 69, and 74, and *n* = 3 for gSCEC, specifically gSCEC 57, 64, and 90) compared to SCECs over 48 h (Figure 4E).

Transcript expression of the tumour suppressor protein p53 was also assessed in light of the differences in cell proliferation and growth observed in glaucomatous SCECs. A decrease in p53 mRNA transcript expression of 0.8 (-1.2, 2.7) fold change, (*p* = 0.3753, unpaired *t*-test, *n* = 5 for SCEC, specifically SCEC 71, 91a, 89, 69, and 74, and *n* = 3 for gSCEC, specifically 57, 64, and 90) was observed in gSCECs, compared to SCECs, but this was not statistically significant (Figure 4F). Excluding outlier SCEC 91a, non-significant reduction in p53 transcript expression remained (*p* = 0.2169).

In EndMT, cells shift their source of energy to glycolysis; thus, we measured the mitochondrial function of SCECs using the Agilent Seahorse XF Cell mitochondrial stress test [34]. This test measures mitochondrial function in cells by measuring the oxygen consumption rate (OCR) of these cells. The difference between maximal and basal levels of respiration was measured and referred to as the spare respiratory capacity. The spare respiratory capacity and ATP production rates were calculated for each group and displayed below (Figure 4G and H, respectively). There was no significant difference in ATP production of 1.3 (−0.1, 2.7) pmol/min (*p* = 0.0614, unpaired *t*-test, *n* = 3 for SCEC, specifically SCEC 89, 91a, and 74, and *n* = 2 for gSCEC, specifically gSCEC 57 and 90) observed in gSCECs, compared to SCECs (Figure 4G). However, there was a significant decrease in spare respiratory capacity of 51.3 (35.2, 67.5) pmol/min (*p* = 0.0021, unpaired *t*-test, *n* = 3 for SCEC, specifically SCEC 89, 91a, and 74, and *n* = 2 for gSCEC, specifically gSCEC 57 and 90) observed in gSCECs, compared to SCECs (Figure 4H).

## 3. Discussion

Our data demonstrate that glaucomatous SCECs are different in many ways, compared to healthy SCECs, showing many characteristics of EndMT. We observed significantly increased protein expression of fibrotic markers α-SMA, collagen I-α1, and fibronectin, possibly driven by increases in the pro-fibrotic marker TGF-β2 observed in gSCECs, compared to SCECs, as well as significantly increased protein expression of endothelial cell marker VE-cadherin. A significant increase in cell size, proliferation, migration, and reduced mitochondrial activity was also observed in gSCECs, compared to SCECs. Taken together, these gSCECs no longer express the healthy phenotype associated with SCECs but appear to express fibrotic characteristics. A summary of the significant differences observed between gSCECs and SCECs is displayed in Table 1.

Glaucoma is an age-related disease, and therefore, gSCECs are typically from older patient cohorts. However, to ensure age was not a confounding factor within SCEC samples, data were reanalysed excluding results from SCEC 74. This donor brings the mean age for the cohort down considerably (Table 2). Statistical significance remained in all assays, with the exception of TGF-β2 transcript expression. Statistical significance remained in TGF-β2 protein expression, however. We, therefore, chose to keep SCEC 74 data included in the manuscript to increase the N for the healthy group. To further show that our results are not solely age dependent, patient samples SCEC 89 and gSCEC 90, with similar ages of 68 and 71 years, respectively, are shown in Figure 3, in which an increase in cell size, proliferation, and α-SMA expression was apparent.

The role of EndMT has been shown to contribute towards the progression of fibrosis of the heart, kidneys, and lungs, as well as cancer progression [35,36,37,38]. It has been shown that EndMT results in the loss of endothelial-specific markers and instead a fibrotic or mesenchymal phenotype is observed, with increased expression of fibrotic markers, such as α-SMA [36,37,38,39]. In this study, however, these cells did not appear to be “losing” their endothelial cell characteristics but instead showed increased expression of endothelial marker VE-cadherin and fibrotic cell markers α-SMA, collagen I-α1, and fibronectin, possibly driven by increases in the pro-fibrotic marker TGF-β2.

Elevated levels of TGF-β observed in glaucomatous tissues have the potential to lead to pro-fibrotic pathway activation in both TM and lamina cribrosa (LC) [40]. In glaucoma, fibrosis is known to occur as a result of alterations to the ECM in the TM, LC, and optic nerve head and plays an important role in the progression of this disease [41]. It is therefore likely that fibrosis is also occurring at the inner wall of SC. Fibrotic marker expression was therefore quantified and compared in both SCECs and gSCECs. A significant fold increase in fibrotic markers α-SMA, collagen I-α1, and fibronectin protein expression of 2.5, 4.7, and 2.3, respectively, was observed in gSCECs, compared to SCECs. A significant fold increase in pro-fibrotic marker TGF-β2 protein expression of 1.7 was also observed. The data in this in vitro study, therefore, suggest that these pro-fibrotic EndMT changes could also be occurring at the inner wall of SC in glaucoma.

VE-cadherin is an important cell–cell adhesion molecule of the adherens junctions in endothelial cells, and disruption of VE-cadherin-mediated cell adhesion results in increased cell permeability [42]. In turn, it has been shown that increased VE-cadherin expression in SCECs coincides with increased transendothelial electrical resistance (TEER) values, following reduction of growth factors in culture [43]. A significant fold increase in VE-cadherin protein expression of 2.5 was observed in gSCECs, compared to SCECs in this study. We hypothesise that increased expression of VE-cadherin in gSCECs contributes to increased outflow resistance in glaucoma due to decreased paracellular permeability through “border pores”. Likewise, glaucomatous SCECs show increased cytoskeletal stiffness, resulting in reduced “intracellular pore” formation, which increases AH outflow resistance and IOP in glaucoma patients [6,44]. However, further study will be required to assess VE-cadherin localisation in gSCECs. An increase in vinculin protein expression has also been observed in SC cells grown on stiffer gel substrates, suggestive of elevated cell stiffness [6]. We observed a fold increase in vinculin protein expression of 0.3 in gSCECs, compared to SCECs, but this was not statistically significant. The increased stiffness of SCECs in glaucoma could be associated with the increased expression of cell adhesion marker VE-cadherin, as well as increased expression of ECM fibrotic markers α-SMA, collagen I-α1 and fibronectin, driven by the pro-fibrotic marker TGF-β2, as observed in this in vitro study.

Rho kinase (ROCK) regulates cell adhesion, migration, and proliferation through control of the actin cytoskeleton and cell contraction [45]. Endothelial dysfunction is associated with a number of vascular diseases and abnormal activation of the ROCK pathway has contributed to disease pathology [46]. Inhibition of the ROCK pathway can prevent endothelial dysfunction in a number of pathological conditions [46]. Since gSCECs appear to be expressing a more fibrotic phenotype and dysfunction of healthy endothelial characteristics, it is possible that ROCK inhibitors could be a beneficial treatment. Thus, the Y-27632 ROCK inhibitor has been shown to significantly reduce actin stress fibres in the cytoskeleton of both TM and SC cell cultures [47]. Moreover, a recent study by Li et al. (2021) reported a significant reduction in IOP in patients with ocular hypertension (OHT) and in a cohort of glucocorticoid (GC)-induced mouse model of OHT, following administration with netarsudil ROCK inhibitor [48]. Treatment with netarsudil also decreased the expression of fibrotic markers α-SMA and fibronectin, showing anti-fibrotic properties capable of restoring outflow facility function and reducing IOP in these mice [48]. This treatment shows promise for potential use as an anti-fibrotic glaucoma therapy. A number of ROCK inhibitors are currently undergoing clinical trial and the potential for these drugs in future glaucoma therapies is clearly evident.

It is clear from the results presented above that gSCECs differ in many ways from healthy SCECs in vitro. gSCECs used in this study were larger in size, proliferated, migrated, and divided at a significantly higher rate, compared to SCECs. Bylsma et al. (1988) showed that healthy SCECs proliferate in human donor tissue, and therefore, proliferation of SCECs likely occurs routinely to repair damages to the inner wall due to mechanical strain and injury of cells [49]. However, in our study, gSCECs proliferated nearly twice as fast as healthy SCECs, despite their large size. This phenomenon has been observed in pulmonary arterial hypertension (PAH), where endothelial dysfunction is described as the main characteristic of early PAH and results in various changes to healthy endothelial cells [50]. These changes include proliferation, migration, EndMT, ECM stiffening, and inflammation due to the Warburg effect and mitochondrial fission [50]. The Warburg effect describes how cancer cells have the potential to carry out mitochondrial respiration through aerobic glycolysis rather than aerobic oxidation [51]. The pulmonary vasculature has been described as being non-compliant due to fibrosis and stiffening, alongside “a cancer-like increase in cell proliferation” [52]. SCECs from glaucoma patients seem to be behaving in a similar manner to these cells affected by PAH. The increase in cell size observed suggests the potential remodelling of these gSCECs. Using a scratch test to model mechanically induced inner wall disruption, gSCECs showed a significantly greater reduction in the scratched area relative to healthy SCECs. This was consistent with their elevated doubling time. There are potentially some epigenetic changes occurring in these gSCECs, resulting in increased proliferation and migration, which warrants further study. p53 mRNA encodes for a tumour suppressor protein that is involved in various cellular processes including cell division, cell cycle control, and apoptosis [53]. We did not detect a significant change in p53 transcript levels, which may suggest that it is not involved in the observed increased proliferation in gSCECs in this study. However, it is well documented that p53 regulation can occur at the post-translational level [54]; thus, analysis of p53 protein modifications and other cell cycle genes, such as Cyclin-A2 (CCNA2) and p16, could be the focus of future studies to provide insight on this mechanism.

gSCECs also showed a significant reduction in mitochondrial activity, compared to SCECs. Mitochondrial spare respiratory capacity in gSCECs was less than half of what was observed in SCECs, while ATP production was unchanged between SCECs and gSCECs. Glaucomatous LC cells have also displayed significantly decreased spare respiratory capacity capabilities, compared to healthy LC cells, as well as producing an equivalent amount of ATP [55]. These glaucomatous LC cells produce significantly less ADP when supplied with either glucose or galactose [55]. This study by Kamel et al. (2020) demonstrated evidence of the Warburg effect in glaucomatous LC cells, as the expression of glycolytic markers was elevated in glaucoma cells at both a transcript and protein level [55]. Increased proliferation has also been observed in glaucomatous LC cells [56]. It is, therefore, possible that gSCECs could be mimicking this phenomenon observed in glaucomatous LC cells, as they both show increased profibrotic markers, cell proliferation, as well as decreased mitochondrial respiration capabilities. Future studies may provide interesting insight into the glycolytic pathway of mitochondrial respiration in gSCECs.

In conclusion, gSCECs used in this study appear to have undergone a range of morphological and molecular changes, compared to SCECs from healthy individuals. It is apparent that gSCECs deviate from and no longer represent healthy endothelial cells but appear to be more fibrotic or mesenchymal in phenotype.

## 4. Materials and Methods

### 4.1. Cell Culture

This study was performed in primary cell cultures. Human SCECs were isolated, cultured, and characterised according to previous protocols at Duke University [43,57,58]. Cells were isolated from the SC lumen of human donor eyes using a cannulation technique. These isolated cells were then tested for positive expression of VE-cadherin and fibulin-2 but an absence of myocilin induction upon treatment with 100 nM of dexamethasone for 5 days. Confluent cells displayed a characteristic linear fusiform morphology, were contact inhibited, and generated a net TEER value greater than 10 Ω.cm^2^. SCECs used in this study can be found in Table 2. The specific cell lines used for each experiment are indicated in the Results. While SCEC 74 greatly reduces the mean age for the control cohort, excluding this sample from the analysis does not affect the outcomes of the results (apart from TGF-β2 transcript expression) or the overarching hypotheses. All SCECs were used between passages 4 and 6. SCECs were grown to confluence for one week in low glucose Dulbecco’s modified Eagle medium (Gibco, Life Sciences, Grand Island, NY, USA) supplemented with 10% performance plus foetal bovine serum (FBS) (Gibco, Life Sciences) and 1% penicillin/streptomycin/glutamine (Gibco, Life Sciences) in a 5% CO2 incubator at 37 °C. Cultured cells were passaged with trypsin-EDTA (Gibco-BRL) to maintain exponential growth.

### 4.2. Quantitative Real-Time PCR

Both SCECs and gSCECs were seeded at 3 × 10^5^/well on a 6-well plate and fed every second day until the cells formed a confluent monolayer. Total RNA was then extracted using RNEasy Mini Kit (Qiagen, MD, USA) according to the manufacturer’s protocol. Genomic DNA contamination was eliminated by DNase treatment. The RNA concentration of each sample was quantified using a NanoDrop^®^ Spectrophotometer ND-100 and equal concentrations were reverse-transcribed into cDNA using High-Capacity cDNA Reverse Transcription Kit (Applied Biosystems, Waltham, MA, USA). SensiFAST SYBR Hi-ROX Kit (Bioline, London, UK) was used according to the manufacturer’s protocol, along with primer pairs, and loaded onto a 96-well plate (Applied Biosystems). The plate was run on a StepOnePlus Real-Time PCR system (Applied Biosystems). Primer pair sequences can be found below in Table 3. The Threshold cycle (Ct) values of samples from both healthy and glaucomatous SCEC were determined, and averages were calculated. The mean normalised expression (∆Ct) of RNA encoding both fibrotic and endothelial cell markers was determined and analysed. Normalised gene expression was calculated by using the equation ∆Ct = Ct (gene of interest) − Ct (housekeeping genes). Expression levels of each sample were normalised against β-actin or GAPDH housekeeping genes. ∆∆CT was calculated by subtracting the sample of interest from a control sample, a healthy SCEC sample which is kept constant as the control sample throughout ∆∆CT = ∆CT (sample of interest) − ∆CT (control sample). The 2-(∆∆CT) method was then used to calculate relative fold gene expression for each sample.

### 4.3. Western Blot

Both SCECs and gSCECs were seeded at 3 × 10^5^/well on a 6-well plate and fed every second day until the cells formed a confluent monolayer. Protein lysates were isolated from this confluent monolayer of cells using a protein lysis buffer containing 50 mM Tris pH 7.5, 150 mM NaCl, 1% NP-40, 1% SDS (working solution) and kept at 4 °C. 1X protease inhibitor cocktail (Roche, Basel Switzerland) is added to 10 mL of this protein lysis buffer and dissolved in solution. The homogenate was centrifuged at 9300 RCF (IEC Micromax microcentrifuge, 851 rotor) at 4 °C for 20 min and the supernatant was stored at −80 °C until needed. Protein concentration was determined by BCA Protein Assay Kit (Pierce, IL, USA) with bovine serum albumin (BSA) at 2 mg/mL as standards on 96-well plates according to the manufacturer’s protocol. Then, 30–50 μg of total protein was loaded in each lane. Protein samples were separated by electrophoresis on 5–10% SDS–PAGE under reducing conditions and electro-transferred to PVDF membranes. After blocking with 5% blotting grade blocker non-fat dry milk in TBS for 1 h at room temperature, membranes were incubated overnight at 4 °C with the following primary antibodies: rabbit polyclonal to alpha-smooth muscle actin (ab5694; dilution of 1:500); rabbit polyclonal to VE-cadherin (ab33168; dilution of 1:1000); rabbit polyclonal to vinculin (ab155120; dilution of 1:1000); mouse monoclonal to transforming growth factor beta-2 (ab36995; dilution of 1:1000); rabbit monoclonal to von Willebrand factor (ab154193; dilution of 1:500); rabbit polyclonal to collagen I (ab34710; dilution of 1:1000); rabbit polyclonal to fibronectin (ab2413; dilution of 1:1000). Blots were washed with TBS-Tween and incubated with either horseradish peroxidase-conjugated polyclonal rabbit IgG secondary antibody (Sigma AB154; dilution of 1:2000) or anti-mouse IgG peroxidase secondary antibody produced in rabbit (Sigma A9044; dilution of 1:10,000). The blots were developed using an enhanced chemiluminescent kit (Pierce Chemical Co., Dallas, TX, USA) and imaged on the C-DiGit^®^ Blot Scanner. Each blot was probed with rabbit polyclonal to GAPDH (ab9485; dilution of 1:1000) as a loading control. Protein band intensities were quantified and analysed using Image J (Version 1.50c). The percentage reduction in band intensity was calculated relative to a healthy SCEC sample, which was standardised to represent 100% and normalised against GAPDH. Results are presented as gSCEC samples relative to healthy SCEC samples.

### 4.4. Immunocytochemistry

Both SCECs and gSCECs were seeded at 1 × 10^4^/well and grown on Lab-Tek II 8-well chamber slides. Briefly, 72 h post seeding, once the cells formed a confluent monolayer on the bottom of the chamber slide, these cells were fixed using 4% paraformaldehyde (pH 7.4) for 20 min at room temperature and then washed 3 times in PBS for 5 min. Cell monolayers were blocked in PBS containing 5% normal goat serum and 0.1% Triton X-100 at room temperature for 1 h. α-SMA primary antibody was made up in blocking buffer (1:500; Abcam) and incubated overnight at 4 °C. Goat anti-rabbit secondary was also made up in blocking buffer (1:500; Abcam), along with phalloidin (1:3000; Invitrogen) and incubated for 2 h at room temperature in a humidity chamber. Following incubation, chamber slides were mounted with Aqua-Polymount (Polyscience, Warrington, PA, USA) after nuclei-counterstaining with DAPI at 1:5000 for 60 s. Fluorescent images of SCEC monolayers were captured using a confocal microscope (Zeiss LSM 710) and processed using imaging software ZEN 2012.

### 4.5. Cell Morphology

Both SCECs and gSCECs were seeded at a lower density of 5 × 10^3^/well, compared to cells used for ICC in order to measure individual cells and not a confluent monolayer of cells. These cells were seeded on Lab-Tek II 8-well chamber slides. Fluorescent images of individual cells, both healthy and glaucomatous SCEC, were captured using a confocal microscope at 40× (Zeiss LSM 710) and processed using imaging software ZEN 2012. Five measurements of both height and width of each cell were taken. These measurements were then averaged for each cell and the area (μm^2^) calculated using the ellipse formula *(A = πab),* where a = height and b = width. This was then analysed using GraphPad Prism 6.0.

### 4.6. Cell Proliferation (MTS) Assay

Both SCECs and gSCECs were seeded at 1 × 10^5^/well on a 12-well plate. These cells were left for 24 h, and a cell proliferation (MTS) assay was then carried out. CellTitre 96 Aqueous One Solution Reagent (Promega, Madison, WI, USA) was thawed and mixed with culture medium at a 1:5 dilution. Cells were incubated with this mixture for a period of 2 h, before transferring the media to a fresh 96-well plate. The absorbance of each well was recorded at 450 nm on a spectrophotometer (Multiskan FC, Thermo Scientific, Waltham, MA, USA). After blanking against wells with reagent and no cells, gSCECs were presented relative to healthy SCECs.

### 4.7. Migration Scratch Assay

Both SCECs and gSCECs were seeded at 1 × 10^4^/well on a 24-well plate. Once cells reached ~90% confluency, a P20 tip was used to scratch a line down the centre of each well, leaving a control well for each cell line unscratched. This in vitro scratch test has been identified as a convenient and inexpensive method for analysing cell migration [59]. Photographs of each well were taken using confocal microscopy at various time points until the cells migrated across the scratch line, and there was no longer a gap visible. These pictures were processed using imaging software ZEN 2012. The reduction of the scratch line was calculated up to 48 h post scratch, and the relative reduction in the scratch line was calculated.

### 4.8. Seahorse Assay

The Agilent Seahorse XF Cell Mito Stress Test was used to measure mitochondrial function by directly measuring the oxygen consumption rate (OCR) of cells on the Seahorse XFe96 Analyser. Respiration assays using the Seahorse XF analysers have been performed by a number of investigators using various cell types [34,60,61,62]. The difference between maximal and basal levels of respiration is measured and referred to as the spare respiratory capacity. It is a plate-based live-cell assay that allows the measurement of mitochondrial function in real time. The assay uses the built-in injection ports on XF sensor cartridges to add modulators of respiration into the cell well during the assay to reveal the key parameters of mitochondrial function. The modulators included in this assay kit are oligomycin, carbonyl cyanide-4 (trifluoromethoxy) phenylhydrazone (FCCP), rotenone, and antimycin. Both SCECs and gSCECs were resuspended in galactose media to force them to use their mitochondria and seeded onto a Seahorse 96-well plate (Agilent) at a concentration of 1 × 10^4^/well, 24 h prior to the assay. The Seahorse cartridge was hydrated in calibrant and incubated in a non-CO^2^ incubator at 37 °C overnight. On the day of the assay, all of the compounds above are added to XF Base Medium and loaded into the cartridge. Two different concentrations of FCCP were used (0.4 μm and 0.8 μm) for this cell type, to determine which was more suitable. Results showed 0.8 μm was more suitable. Culture media was substituted for Seahorse XF DMEM medium, pH7.4 (Agilent, Santa Clara, CA, USA), supplemented with 1 mM sodium pyruvate, 2 mM L-glutamine, and 10 mM galactose, and placed in a non-CO^2^ incubator at 37 °C 1 h prior to the reaction. The assay was then run on a Seahorse XFe96 Analyser. A Bradford assay (Thermo Fisher Scientific, Waltham, MA, USA) was later carried out to determine protein concentration, and data from the Seahorse assay were then normalised to this concentration.

### 4.9. Statistical Analysis

For real-time PCR, Western blot analysis, and functional assay experiments, unpaired Student’s *t*-tests were carried out using GraphPad Prism 6.0, unless otherwise stated. Outliers were identified using the ROUT method, where Q = 1%, on GraphPad Prism 6.0. Statistical significance was indicated by *p* ≤ 0.05.

## Figures and Tables

**Figure 1 ijms-22-09446-f001:**
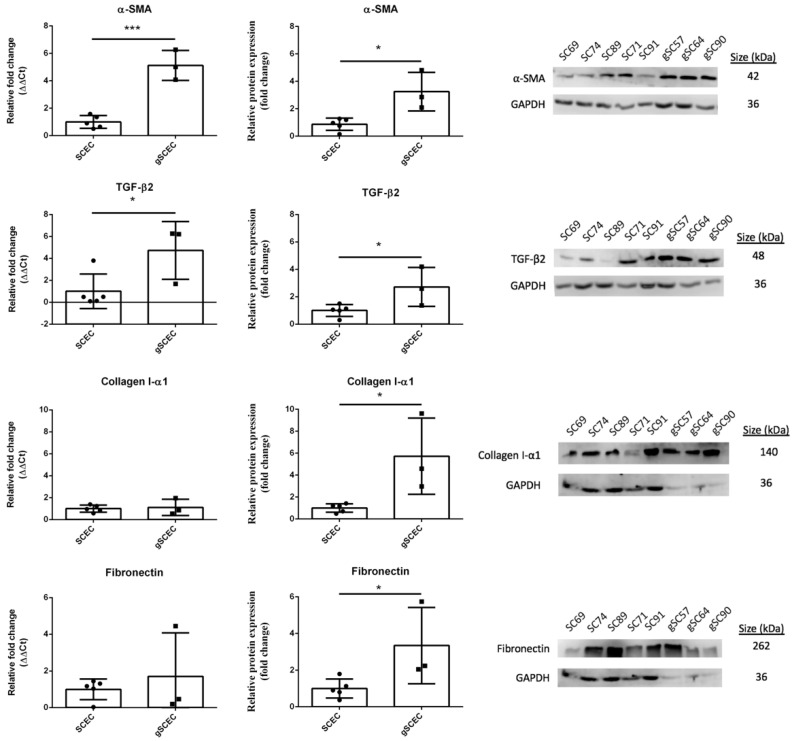
Relative expression of α-SMA, TGF-β2, collagen I-α1 and fibronectin in heathy versus glaucomatous SCECs. Shown are mRNA transcript (left column) and protein (middle column) quantification of the fibrotic markers, α-SMA, collagen I-α1, and fibronectin, as well as pro-fibrotic cytokine TGF-β2 in gSCECs relative to healthy SCECs. Representative blots are shown in the right columns. A significant increase in α-SMA transcript (*p* = 0.0003) and protein (*p* = 0.0105) expression was observed between gSCECs and SCECs. A significant increase in TGF-β2 transcript (*p* = 0.0431) and protein (*p* = 0.0389) expression was also observed between gSCECs and SCECs. While no significant difference in collagen I-α1 and fibronectin transcript expression was observed (*p* > 0.05), a significant increase in both collagen I-α1 (*p* = 0.0190) and fibronectin (*p* = 0.0457) protein expression was observed in gSCECs, compared to SCECs. Mean +/− SD is shown above with each data point representing an individual biological sample, *n* = 5 for SCEC and *n* = 3 for gSCEC. Statistical analysis was carried out using an unpaired Student’s *t*-test and statistical significance was indicated by the following * = (*p* < 0.05) and *** = (*p* < 0.001).

**Figure 2 ijms-22-09446-f002:**
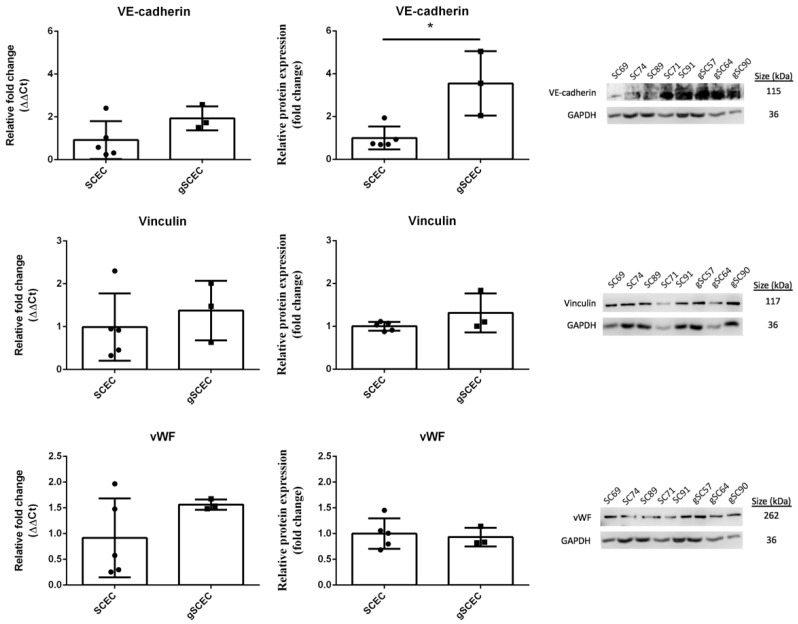
Comparison of endothelial marker expression in healthy versus glaucomatous SCECs. Shown are mRNA transcript (left column) and protein (middle column) quantification of the endothelial markers VE-cadherin, vinculin and vWF in gSCECs relative to healthy SCECs. Representative blots are shown in right columns. A significant increase in VE-cadherin protein (*p* = 0115) expression was observed in gSCECs, compared to SCECs. No significant increase in vinculin and vWF transcript (*p* > 0.05) and protein (*p* > 0.05) expression was observed. Mean +/− SD is shown above with each data point representing an individual biological sample, *n* = 5 for SCEC and *n* = 3 for gSCEC. Statistical analysis was carried out using an unpaired Student’s *t*-test and statistical significance was indicated by * = (*p* < 0.05).

**Figure 3 ijms-22-09446-f003:**
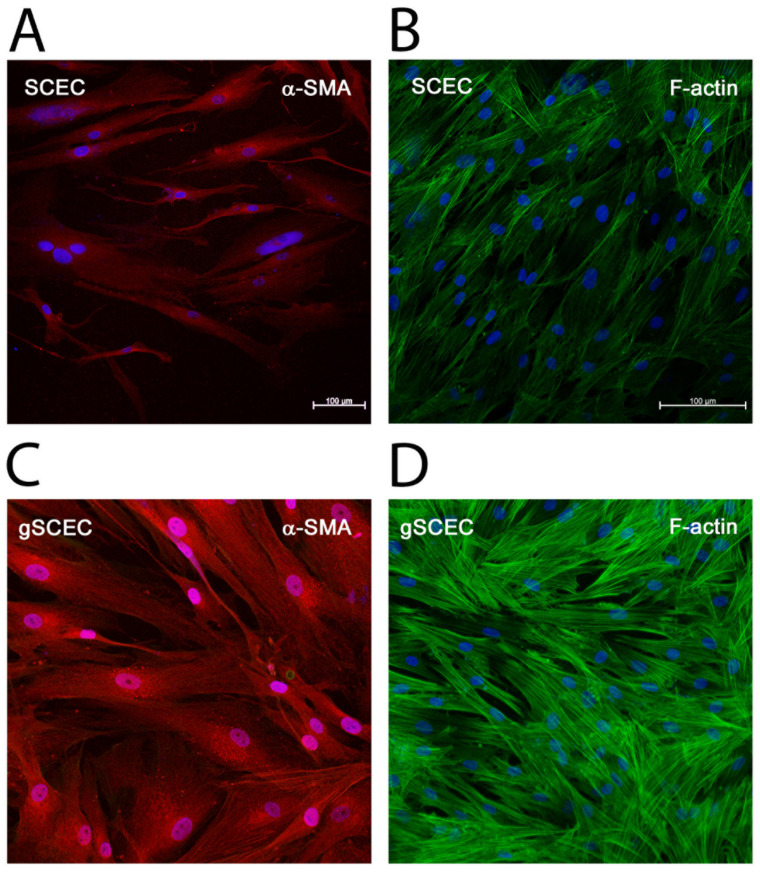
Elevated levels of α-SMA and F-actin protein in glaucomatous SCECs, compared to healthy SCECs. Increased expressions of cytoskeletal proteins α-SMA (red) and F-actin (green) were observed in gSCECs (**C** + **D**), compared to healthy SCECs (**A** + **B**). gSCECs appeared to be larger in size and showed increased expression of α-SMA (**C**), compared to healthy SCECs (**A**). An increase in actin stress fibres and the density of F-actin cytoskeletal protein expression was apparent in gSCECs (**D**), compared to healthy SCECs (**B**). These cells were seeded at the same time and at the same density; however, gSCECs appeared more confluent (**C** + **D**) than healthy SCECs (**A** + **B**). DAPI is represented in blue. Scale bar represents 100 μm. SCEC 89 and gSCEC 90 above represent *n* = 3 cell strains analysed for both gSCECs and healthy SCECs.

**Figure 4 ijms-22-09446-f004:**
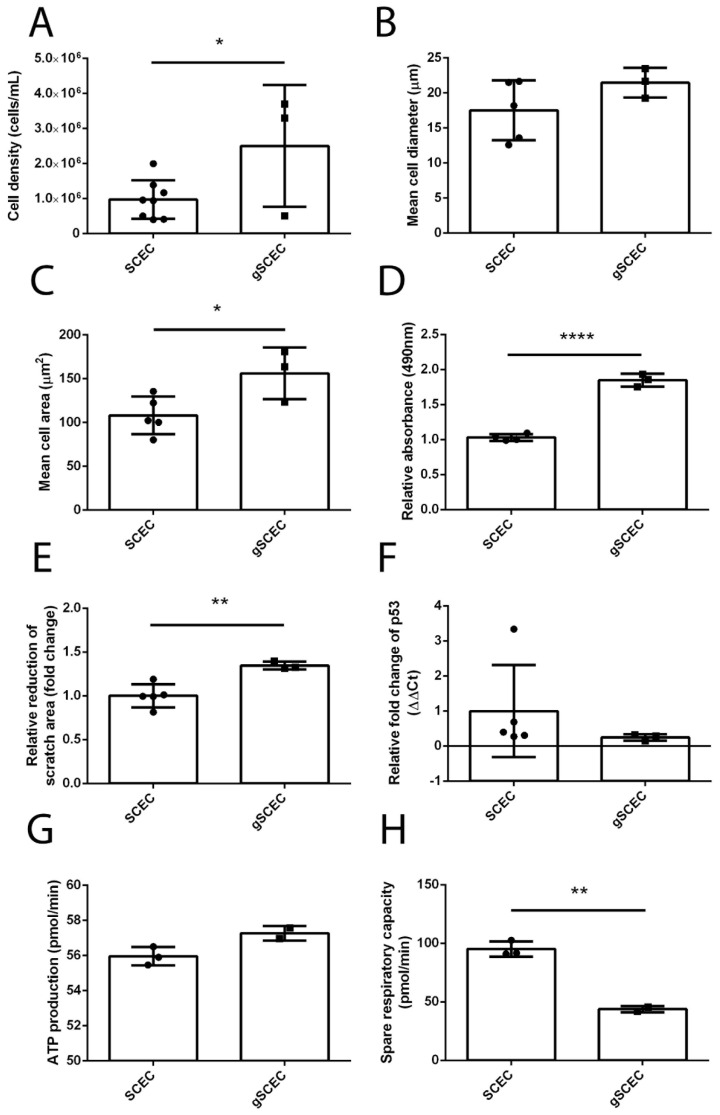
Increased cell size, proliferation, migration, and reduced mitochondrial activity observed in gSCECs, compared to heathy SCECs. A significant increase in cell density (*p* = 0.0417, *n* = 8 for SCEC and *n* = 3 for gSCEC) was observed in gSCECs, compared to SCECs (**A**). A non-significant difference in the mean cell diameter of these seeded cells (*p* = 0.1941, *n* = 5 for SCEC and *n* = 3 for gSCEC) was observed in gSCECs, compared to SCECs using the LUNA Automated Cell Counter (**B**). Using a separate method, a significant increase in mean cell area (*p* = 0.0361, *n* = 5 for SCEC and *n* = 3 for gSCEC) was observed in gSCECs, compared to SCECs (**C**). A significant increase in cell proliferation (*p* < 0.0001, *n* = 4 SCEC and *n* = 3 for gSCEC) (**D**) and cell migration (*p* = 0.0054, *n* = 5 for SCEC and *n* = 3 for gSCEC) (**E**) was also observed in gSCECs, compared to SCECs. No significant difference in relative transcript expression of tumour suppressor p53 was observed between gSCECs and SCECs (*p* = 0.3753, *n* = 5 for SCEC and *n* = 3 for gSCEC) (**F**). There was also no significant difference in ATP production observed between SCECs and gSCECs (*p* = 0.0614, *n* = 3 for SCEC and *n* = 2 for gSCEC) (**G**). A significant reduction in spare respiratory capacity was observed, however, in gSCECs, compared to SCECs (*p* = 0.0021, *n* = 3 for SCEC and *n* = 2 for gSCEC) (**H**). Mean ± SD is shown above, and statistical analysis was carried out using an unpaired Student’s *t*-test for all. Statistical significance was indicated by the following * = (*p* < 0.05), ** = (*p* < 0.01) and **** = (*p* < 0.0001).

**Table 1 ijms-22-09446-t001:** Summary of significant changes observed between gSCECs and SCECs. This table summarises the statistically significant changes observed in gSCECs, compared to SCECs. ↑ represents an increase or upregulation observed in gSCECs, compared to SCECs. ↓ represents a decrease or downregulation observed in gSCECs, compared to SCECs. Statistical significance is indicated by the following * = (*p* < 0.05), ** = (*p* < 0.01), *** = (*p* < 0.001), and **** = (*p* < 0.0001).

Assay	Significant Changes
α-SMA transcript expression	↑ ***
α-SMA protein expression	↑ *
TGF-β2 transcript expression	↑ *
TGF-β2 protein expression	↑ *
Collagen I-α1 protein expression	↑ *
Fibronectin protein expression	↑ *
VE-cadherin protein expression	↑ *
Cell density	↑ *
Cell area (ICC)	↑ *
Cell migration	↑ **
Cell proliferation	↑ ****
Spare respiratory capacity	↓ **

**Table 2 ijms-22-09446-t002:** Donor information for both healthy and glaucomatous SCEC strains.

Group	Cell Strain No.	Age (Years)	Sex
Healthy	SCEC 71	44	Male
	SCEC 91a	74	Female
	SCEC 89	68	Male
	SCEC 69	45	Male
	SCEC 74	0.7	Male
	SCEC 68	30	Unknown
	SCEC 81	75	Unknown
	SCEC 86	62	Male
Glaucoma	gSCEC 57	78	Male
	gSCEC 64	78	Male
	gSCEC 90	71	Female

**Table 3 ijms-22-09446-t003:** Primer pair sequences used in quantitative real-time PCR.

Primer Pair	Forward Sequence (5’ to 3’)	Reverse Sequence (5’ to 3’)
α-SMA	CCGACGAATGCAGAAGGA	ACAGAGTATTTGCGCTCCGAA
TGF-β2	ACGGATTGAGCTATATCAGATTCTCA	TGCAGCAGGGACAGTGTAAG
Collagen Iα-1	TTTGGATGGTGCCAAGGGAG	CACCATCATTTCCACGAGCA
Fibronectin	AGCGGACCTACCTAGGCAAT	GGTTTGCGATGGTACAGCTT
VE-cadherin	GCACCAGTTTGGCCAATATA	GGGTTTTTGCATAATAAGCAGG
Vinculin	TGGACGAAGCCATTGATACCA	AGCTCTTTTGCAGTCCAGGG
vWF	GTTCGTCCTGGAAGGATCGG	CACTGACACCTGAGTGAGAC
β-actin	GGGAAATCGTGCGTGACAT	GTGATGACCTGGCCGTAG
GAPDH	TGTAGTTGAGGTCAATGAAGGG	ACATCGCTCAGACACCATG

## Data Availability

The data presented in this study are available on request from the corresponding author.

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
