# Peer review of "Fibrotic Changes to Schlemm’s Canal Endothelial Cells in Glaucoma"

_ijms, 2021, doi:10.3390/ijms22179446_

Round 1
Reviewer 1 Report
Congratulation to your research which describes possible pathogenesis of Schlemm's canal and TM changes in glaucoma. It suggests possible way for the innovative treatment. I recommend to continue and to study more specimens from different types of glaucoma.
Author Response
We thank the reviewer for their comments. We hope to continue with this study.
Reviewer 2 Report
primary cultures of Schlemm’s canal endothelial cells (SCECs) from glaucoma patients and normal individuals. The data suggest increased fibrosis, cell proliferation and migration and decreased respiratory capacity of glaucomatous compared to normal SCECs. This study adds to the current understanding of glaucoma pathophysiology, particularly of the aqueous humor outflow pathway. However, there are some concerns about interpretation of the data and some of the conclusions drawn.
Comments
The mean ages of donors of normal and glaucomatous SCECs used in this study are 46.34 and 75.66 years, respectively. Could the observed differences in gene expression, cell morphology, proliferation, migration and respiration be due to the difference in age between normal and glaucomatous SCECs? How was the effect of age distinguished from that of the disease?
The authors hypothesize that increased VE-cadherin expression in glaucomatous SCECs contributes to increased outflow resistance in glaucoma. These SCECs have a mesenchymal rather than an endothelial morphology in culture (Figure 3). What is the sub-cellular localisation of VE-cadherin in these cells? Is the localization consistent with the proposed hypothesis?
The authors also suggest that glaucomatous SCECs produce ATP through aerobic glycolysis rather than oxidative phosphorylation due to the Warburg effect as has been reported in primary cultures of lamina cribrosa cells (ref#51). This might be the case but in the absence of supporting data this conclusion is speculative and hence avoided.
The data does not support the conclusion, “A decrease in p53 mRNA transcript expression observed in gSCECs could explain why these cells were growing and proliferating at a much higher rate compared to SCECs.” because the difference in p53 expression levels between glaucomatous and normal SCECs is not significant.
Can the author clarify in the Abstract and at the beginning of Results that the study was performed in primary cell cultures?
The description of some of the results, especially non-significant results is a bit misleading. The authors refer to trends that are not informative because of high within group variability. When a difference was not significant this should be clearly stated. For example, in Figure 2, there is no difference in expression of endothelial markers vinculin and van Willebrand factor at the transcript and protein levels between glaucomatous and normal SCEC but VE-cadherin protein is upregulated in the diseased SCEC. Similarly, in section 2.4, lines 176-178 and lines 193-197, there is no significant difference in cell size and levels of p53 mRNA,
respectively, between normal and glaucomatous SCECs. Figure 4 legend also accordingly
needs correction.
How reproducible are the data shown in Figure 3? Please include scale bars in Figure 3C and
3D to demonstrate that cells in these panels were imaged at the same magnification as in
3A and 3B.?
Materials and Methods-
There is reference to “treated samples”. What were samples treated with?
Is the indicated composition of protein lysis buffer of a working or stock buffer?
Centrifugation is indicated as centrifugal force rather than rpm.
Please include specific details of the primary and secondary antibodies used, including hot
species, dilutions and catalogue numbers.
Considerable variation in writing style, including usage of tense (in Result) was noted in the
manuscript. Could the authors adopt a consistent writing style?
The Discussion is a bit repetitive. Could it be more concise?
Lines 291-293, “Using a scratch test, to model mechanically induced inner wall disruption,
gSCECs showed a significant increase of 0.4 fold on reduction of the scratched area relative
to healthy SCECs.” could be “Using a scratch test, to model mechanically induced inner wall
disruption, gSCECs showed a significantly greater reduction in scratched area relative to
healthy SCECs.”?
Please define abbreviations upon first appearance. Eg. JCT and TEER.
Please ensure that references are formatted according to the journal guidelines, particularly
abbreviations of journal names.

Reviewer 3 Report
In their study, the authors compared several characteristics of Schlemm’s canal endothelial cells derived from healthy and glaucomatous subject and find that they show different expressions of several proteins and different morphological characteristics. The study is of interest and of merit, however, some points needs to be addressed, especially concerning the methods.
Abstract
Please provide the origin of the cells/cell model and the methods in the abstract. Which of the changes described were significant? Please indicate.
Methods
Line 331, following. Please give a brief description of the methods how the cells were isolated. How was the identity of the cells confirmed?
Please give the exact density (cell number per well) that was used in the experiments.
Except for the reference to the cell isolation, no references are given in the method secretion. Please include appropriate references.
Results/Figures
The authors claim an elevated level of alpha-SMA and F-actin. Did they adjust for exposure time? Indeed, the differences shown in the figure (3) appear to be more related to exposure time. Please indicate measurements. Also, the scale bar is missing in C and D. Did the author quantify the intensity (normalized for exposure time) of the signal? This could give an objective indication.
Also the differences in cell size measured in LUNA could be confirmed in IF.
Page 6 and figure 4, why are the sample sizes so different between normal and g-SCEC (n=29 vs. n=9)?
Figure 4, please indicate in F which mRNA is measured here (p53).
Discussion
Page 9, second paragraph, it is not quite clear to the reader why rho kinase inhibitors are beneficial, as they have not been introduced before. A mention of their role of rho kinase would be beneficial.
Conclusion, lines 321 -327, this is a discussion, not a conclusion. Please adjust.
Minor
Figures:
The blots in the figure are very small, I would suggest increasing them in size.
Round 2
Reviewer 2 Report
Thank you to the authors for addressing the comments and the revised manuscript. Thank you also for re-analyzing the data. Most of the comments made by this reviewer have been satisfactorily addressed. However, responses to some of the comments require further clarification in the manuscript.
- In response to the comment regarding the potential effect of donor age on the presented data, the authors have presented the results of reanalysis excluding SCEC 74. Please also discuss and include the conclusion of this reanalysis in the Discussion, in the manuscript.
- Table 2 lists donor details of SCEC lines used for gene and protein expression analyses (Figures 1, 2 and probably 4F). The study used a larger number of cell lines for comparison of cell density between glaucomatous and normal SCEC (n=29 SCEC and n=9 gSCEC; Figure 4A), and apparently a sub-set of these lines for other cellular assays (Figure 4B-E and G-H). Hence, donor details of all the cell lines used in the study should be listed in Table 2. Also indicate which lines were used for which experiments. In each assay, if there were any outlier/s in either group then please determine its/their effect on statistical significance of results.
- Lines 313- 316, “A non-significant decrease in p53 mRNA transcript expression was observed in gSCECs compared to SCECs. There are also many other genes associated with cell cycle however, that could explain how and why these gSCECs continued to grow and proliferate at such a high rate.”
Non-significant reduction in p53 expression levels indicates that p53 is not involved in increased proliferation of gSCECs observed in this study. However, other cell cycle genes could be involved. Please clarify this point in the Discussion.
- Lines 344-345, “While SCEC 74 greatly reduces the mean age for the control cohort, excluding this sample from analysis does not affect the outcomes.” of the results or hypotheses.”
This is not entirely true because increase in TGF-β2 transcript levels was not significant after reanalysis.
- Generally, protein lysis buffer has a final concentration of 25-50mM Tris, 150-250 mM NaCl, 1% NP-40 and 1% SDS. Please clarify the basis for the buffer constitution used in the study.
- Please also state the findings of mitochondrial function assay in the Abstract.
Reviewer 3 Report
The authors have answered all comments to the reviewer's satisfaction.
Author Response
We thank the Reviewer for their comments.